# Stroke-SCORE: Personalizing Acute Ischemic Stroke Treatment to Improve Patient Outcomes

**DOI:** 10.3390/jpm15010018

**Published:** 2025-01-04

**Authors:** Jessica Seetge, Balázs Cséke, Zsófia Nozomi Karádi, Edit Bosnyák, László Szapáry

**Affiliations:** 1Stroke Unit, Department of Neurology, University of Pécs, 7624 Pécs, Hungary; j.seetge@gmx.de (J.S.); karadi.zsofia@pte.hu (Z.N.K.); bosnyak.edit@pte.hu (E.B.); 2Department of Emergency Medicine, University of Pécs, 7624 Pécs, Hungary; cseke.balazs@pte.hu (B.C.)

**Keywords:** acute ischemic stroke, Stroke-SCORE, simplified clinical outcome risk evaluation

## Abstract

**Background/Objectives**: Acute ischemic stroke (AIS) is a leading cause of disability and mortality worldwide. Despite advances in interventions such as thrombolysis (TL) and mechanical thrombectomy (MT), current treatment protocols remain largely standardized, focusing on general eligibility rather than individual patient characteristics. To address this gap, we introduce the Stroke-SCORE (Simplified Clinical Outcome Risk Evaluation), a predictive tool designed to personalize AIS management by providing data-driven, individualized recommendations to optimize treatment strategies and improve patient outcomes. **Methods:** The Stroke-SCORE was derived using retrospective data from 793 AIS patients admitted to the University of Pécs (February 2023–September 2024). Logistic regression analysis identified age, National Institutes of Health Stroke Scale (NIHSS) score at admission, and pre-morbid modified Rankin Scale (pre-mRS) score as key predictors of unfavorable outcomes at 90 days (defined as modified Rankin Scale [mRS] score > 2). Based on these predictors, a simplified risk score was developed to stratify patients into low-, moderate-, and high-risk groups, guiding treatment decisions on TL, MT, combination therapy (TL + MT), or standard care (SC). Internal validation was performed to assess the model’s predictive performance via receiver operating characteristic (ROC) analysis and isotonic regression calibration with bootstrapping. **Results:** The Stroke-SCORE was moderately positively correlated with a 90-day mRS score > 2 (odds ratio [OR] = 0.70, 95% confidence interval [CI]: 0.58–0.83, *p* < 0.001), with an area under the curve (AUC) of 0.86, a sensitivity and specificity of 79% and 81%, respectively, and an overall accuracy of 80%. Simulations indicated that personalized treatment guided by the Stroke-SCORE significantly reduced unfavorable outcomes. **Conclusions:** The Stroke-SCORE demonstrates strong predictive performance as a practical, data-driven approach for personalizing AIS treatment decisions. In the future, external, multicenter prospective validation is needed to confirm its applicability in real-world settings.

## 1. Introduction

Acute ischemic stroke (AIS) remains a major public health concern, affecting more than 13 million people annually and ranking as a leading cause of disability and mortality worldwide [1]. While standard treatment protocols, such as thrombolysis (TL) and mechanical thrombectomy (MT), have significantly improved outcomes for eligible patients [2], they often fail to account for individual variability [3]. As a result, many patients experience suboptimal outcomes, highlighting the need for personalized approaches to AIS treatment.

Current guidelines for TL recommend its administration within 3 h for both younger (≤80 years) and older (>80 years) patients with severe strokes (National Institutes of Health Stroke Scale [NIHSS] > 25) or mild but disabling symptoms. For the 3 to 4.5 h window, TL is primarily recommended for patients under 80 years with NIHSS < 25 and may be reasonable in selected cases with a pre-morbid modified Rankin Scale (pre-mRS) score > 2. Conversely, TL is contraindicated in mild, non-disabling strokes (NIHSS 0–5). Similarly, MT eligibility is determined by factors such as pre-mRS score (0–1), NIHSS score (>6), and infarct location (large vessel occlusion [LVO] of the internal carotid artery [ICA] or proximal middle cerebral artery [MCA]) within the first 6 h. Extended treatment windows up to 16 or 24 h rely on advanced imaging criteria, including DEFUSE-3 and DAWN, to identify patients likely to benefit based on infarct volume and perfusion mismatch [2].

Despite these advancements, only 40–50% of MT-eligible and 50–60% of TL-eligible patients achieve functional independence (modified Rankin Scale [mRS] score of 0–2) at 90 days post-stroke [4,5]. Outcomes are further influenced by individual factors, with older age and higher NIHSS scores at admission consistently associated with worse recovery and higher mRS scores at 90 days [4]. These challenges highlight the need for treatment strategies that incorporate patient-specific characteristics into decision-making to optimize outcomes.

To address this need, the Stroke-SCORE (Simplified Clinical Outcome Risk Evaluation), a decision-support tool designed to provide personalized treatment recommendations for AIS patients, was developed. Using readily available clinical data, the Stroke-SCORE incorporates key factors such as age, NIHSS score at admission, and pre-mRS score to assist clinicians in making more informed decisions. By predicting individual prognosis, the Stroke-SCORE complements existing guidelines, bridging the gap between generalized treatment protocols and the need for individualized care. This novel approach aims to improve outcomes in AIS management and advance the personalization of stroke care.

## 2. Materials and Methods

### 2.1. Study Design and Dataset

This retrospective cohort study was conducted using data from the prospective Transzlációs Idegtudományi Nemzeti Laboratórium (TINL) STROKE registry. Between February 2023 and September 2024, 914 AIS patients were admitted to the Department of Neurology, University of Pécs. After excluding 121 patients with incomplete data (*n* = 105 missing 90-day outcomes, *n* = 11 missing international normalized ratio [INR] values, and *n* = 5 missing plasma-glucose levels), a total of 793 patients’ medical data were analyzed.

To ensure data reliability, the TINL STROKE registry adheres to strict protocols for consistent and accurate data collection. Patient information is systematically recorded by trained neurology staff following standardized criteria. Data integrity is maintained by cross-checking critical clinical variables against original medical records, with any discrepancies resolved through team consensus. Periodic internal audits are conducted to identify and address inconsistencies or missing data, further enhancing the quality of the registry.

The dataset included patient demographics (age and sex), clinical characteristics (pre-mRS scores, baseline NIHSS scores, plasma-glucose levels, INR values at admission, onset-to-door times, and comorbidities such as hypertension and diabetes mellitus), treatment details (type of recanalization therapy: standard care (SC), TL, MT, or combination therapy [TL + MT]), and 90-day outcomes. AIS was defined as “an episode of neurological dysfunction caused by focal cerebral, spinal, or retinal infarction” [6], without evidence of acute intracranial hemorrhage on imaging.

### 2.2. Outcome Assessment

The primary outcome was assessed 90 days after stroke using the mRS, a well-established measure of functional independence. A favorable outcome was defined as an mRS score of 0–2, whereas an mRS score > 2 indicated an unfavorable outcome [7]. Functional outcomes were evaluated during follow-up visits or via structured telephone interviews conducted by trained personnel, based on the simplified mRS questionnaire algorithm as described by Bruno et al. [8].

### 2.3. Development of the Stroke-SCORE

Key predictors of unfavorable outcomes at 90 days were identified using logistic regression analysis and further confirmed with SHapley Additive exPlanations (SHAP) to quantify the contribution of each feature (Table 1). Age, NIHSS score at admission, and pre-mRS score emerged as the most significant predictors, whereas other factors, such as plasma-glucose levels, INR at admission, onset–door time, hypertension, and diabetes mellitus were found to have minimal impact on model performance. The Stroke-SCORE was developed using only the three most impactful predictors—pre-mRS, NIHSS, and age—to achieve a balance between predictive accuracy and practical applicability in real-world clinical settings.

Based on their contribution to predicting unfavorable outcomes, the Stroke-SCORE assigns weighted points to each significant predictor: age ≥ 80 years: +1 point, NIHSS score at admission > 15: +2 points, and pre-mRS score ≥ 3: +1 point.

The cumulative score stratified patients into low-risk (0–1 points, <30% probability of unfavorable outcome), moderate-risk (2–3 points, 30–70% probability of unfavorable outcome), and high-risk (4+ points, >70% probability of unfavorable outcome) categories.

The predictive model was developed using a Gradient Boosting Classifier, selected for its ability to manage complex, nonlinear relationships among variables. This machine learning technique iteratively optimizes predictive performance, making it well suited for heterogeneous data. To ensure robust evaluation, the dataset was divided into training (80%) and testing (20%) subsets, and features were standardized using StandardScaler to improve model stability and convergence. All statistical analyses were performed in Python to ensure replicability and methodological transparency.

The cut-off values for age, NIHSS score, and pre-mRS score were derived from statistical analysis of the initial dataset to maximize predictive accuracy. Internal validation of the Stroke-SCORE was conducted using receiver operating characteristic (ROC) analysis, which demonstrated a strong predictive accuracy with an area under the curve (AUC) of 0.86. Additionally, calibration analysis via isotonic regression with bootstrapping confirmed that the predicted probabilities aligned closely with observed outcomes, ensuring reliability across risk categories.

### 2.4. Implementation of the Stroke-SCORE

The Stroke-SCORE was integrated into an interactive decision-support tool built with Dash, enabling real-time clinical application (Figure 1). This interface automatically classifies patients into risk categories, allowing clinicians to assess prognosis efficiently. Additionally, it provides visualizations of the predicted probability of unfavorable outcomes, demonstrating the potential impact of various treatment options. Through interactive decision-making, the tool offers individualized treatment recommendations aimed at minimizing the risk of unfavorable outcomes.

### 2.5. Ethics Statement

This study was conducted in accordance with the Declaration of Helsinki and approved by the Scientific and Research Ethics Committee of the Medical Research Council of the University of Pécs (RRF-2.3.1-21-2022-00011), date of approval 1 September 2022 (01/09/22).

## 3. Results

### 3.1. Performance of the Stroke-SCORE

The Stroke-SCORE had a moderate positive correlation with 90-day mRS outcomes, (correlation coefficient 0.51, *p* < 0.001) and demonstrated strong predictive accuracy, with an AUC of 0.86 and an overall accuracy of 80%. The model’s performance metrics included a precision of 0.82, recall of 0.79, sensitivity of 0.79, and specificity of 0.81 (Figure 2). These metrics highlight the model’s ability to effectively differentiate between patients likely to achieve favorable versus unfavorable outcomes.

Moreover, calibration analysis confirmed that the model’s predicted probabilities aligned well with the observed outcomes, indicating that the Stroke-SCORE provides reliable risk estimates for patient prognosis (Figure 3).

### 3.2. Patient Characteristics

Among the 793 patients analyzed, the median age was 72 years (interquartile range [IQR]: 63–80), with 49.3% being male. The median NIHSS score at admission was 5 (IQR: 3–10), indicating that most patients experienced strokes of moderate severity. The median pre-stroke mRS score was 0 (IQR: 0–1), suggesting a generally low level of pre-existing disability. The treatment distribution included 181 patients (22.8%) receiving TL, 150 (18.9%) undergoing MT, 63 (7.9%) receiving combination therapy and 399 (50.3%) receiving SC.

To illustrate how patient characteristics influence treatment selection, Figure 4 presents box plots of age, NIHSS score, and pre-mRS score across treatment groups. These visualizations underscore significant differences in patient characteristics across treatment strategies (Figure 4).

### 3.3. Patient Risk Stratification and Treatment Efficacy Across Different Stroke-SCOREs

The majority of patients were classified as low risk (671; 85%), while 113 (14%) were classified as moderate risk, and 9 (1%) were classified as high risk (Figure 5).

Patients in the low-risk group had an overall average 90-day mRS of 2.18 (95% confidence interval [CI]: 2.02–2.34), those in the moderate-risk group had an average of 4.56 (95% CI: 4.24–4.87), and those in the high-risk group had an average of 5.89 (95% CI: 5.63–6.00), highlighting the general prognosis for each risk category. Table 2 summarizes the average 90-day mRS score across different treatment approaches for each Stroke-SCORE. Notably, differences in 90-day mRS across treatments were statistically significant in the low-risk group (*p* < 0.001), borderline significant in the moderate-risk group (*p* = 0.055), and not significant in the high-risk group (*p* = 0.157).

Logistic regression analysis further indicated that low-risk patients (Stroke-SCORE of 0 or 1) experienced the greatest benefit from TL. These patients tended to be younger (average age: 67.1 years vs. 71.9 years) and had lower NIHSS scores (mean 4.9 vs. 7.7) and pre-mRS scores (mean 0.38 vs. 0.75) compared to other treatment groups (*p* < 0.001). Importantly, even after adjusting for these baseline differences, TL was independently associated with a 30% reduction in the likelihood of unfavorable outcomes compared to other treatments (odds ratio [OR] = 0.70, 95% CI: 0.58–0.83, *p <* 0.001).

Patients who received MT and TL + MT exhibited significantly higher NIHSS scores (mean: 11.2 vs. 6.1, *p* < 0.001; mean: 10.7 vs. 6.8, *p* < 0.001, respectively). Additionally, MT patients had significantly higher plasma glucose levels (mean: 8.35 vs. 7.60, *p* = 0.006). In contrast, the TL + MT group had a lower proportion of patients with diabetes (21% vs. 37%, *p* = 0.009) compared to other treatment groups. MT was associated with a 21% reduction in the likelihood of unfavorable outcomes (OR = 0.79, 95% CI: 0.66–0.95, *p* = 0.013), while TL + MT showed a 13% reduction, although this finding was close to the threshold of statistical significance (OR = 0.87, 95% CI: 0.73–1.04, *p* = 0.060).

### 3.4. Simulated Outcomes

The model was used to create a counterfactual analysis that estimated potential 90-day outcomes if different treatments were applied universally to the cohort: TL was most effective for low-risk patients, with a predicted probability of unfavorable outcomes of 41%, within a 95% CI of 41–46%. MT was associated with a greater likelihood of unfavorable outcomes (54%, 95% CI: 51–55%), particularly in high-risk patients. Combination therapy yielded a predicted probability of 51% (95% CI: 51–55%) (Figure 6).

These findings are robust and supported by the calculated CIs obtained through bootstrapping, which effectively account for uncertainty in the modeling process. The initially predicted probabilities (TL: 41%; MT: 54%; TL + MT: 51%) are consistent with the derived CIs, confirming the stability of the results. While minor variations in the initial estimates may occur due to stochastic elements inherent in the modeling algorithm, such as random sampling or initialization, these fluctuations are negligible and do not alter the overall conclusions.

## 4. Discussion

This study establishes the Stroke-SCORE as a reliable tool for risk stratification and outcome prediction in acute stroke patients. Its moderate correlation with 90-day mRS outcomes and strong predictive accuracy underscore its clinical value, while calibration analysis validates its ability to provide accurate risk estimates.

Detailed analysis of patient characteristics revealed significant differences in age, NIHSS scores and pre-mRS scores across treatment groups, highlighting the influence of baseline factors on treatment selection and outcomes. TL was shown to confer the greatest benefit to low-risk patients, whereas higher-risk patients derived limited benefit from advanced interventions such as MT or combination therapy, underscoring the importance of individualized, risk-based therapeutic strategies.

Simulated counterfactual analysis further illustrated treatment efficacy disparities by risk group. For low-risk patients, TL was associated with the lowest probability of unfavorable outcomes, while MT and combination therapy were less effective, particularly in high-risk patients. These findings highlight the critical importance of tailored therapeutic strategies, guided by tools like the Stroke-SCORE, to optimize clinical decision-making and improve patient outcomes.

### 4.1. Navigating High-Risk Treatment Decisions with the Stroke-SCORE

To demonstrate the utility of the Stroke-SCORE model, consider a hypothetical case: an 82-year-old patient presents with a large vessel occlusion within 4.5 h of symptom onset. The patient has an NIHSS score of 18 and a pre-mRS score of 3, resulting in a total Stroke-SCORE of 4, which categorizes the patient as high-risk.

In this scenario, the Stroke-SCORE model predicts only a modest chance of a favorable outcome with TL or MT, with risk reduction estimates of just 0.14% for TL and 1.02% for MT compared with SC. These figures indicate a relatively high risk of an unfavorable outcome, even when aggressive interventions are used. This prediction underscores that despite eligibility for these treatments, the likelihood of substantial functional recovery is limited.

The Stroke-SCORE supports clinicians in navigating the complexity of making treatment decisions. This highlights that while the patient meets eligibility for both TL and MT, the expected improvement in functional outcome may not justify the risks of intervention. By providing such insights, the model facilitates a more informed discussion among the medical team, the patient, and their family, helping them weigh the potential benefits of intervention against the risks, such as procedural complications and the patient’s overall health status.

### 4.2. Comparison with Existing Scoring Systems

In recent years, several scoring systems have been developed to predict outcomes after AIS. Notable examples include the Acute Stroke Registry and Analysis of Lausanne (ASTRAL) score [9], the DRAGON score (Diabetes, Rankin Scale, Age, Glucose, Onset-to-treatment time, and NIHSS score) [10], and the Totaled Health Risks in Vascular Events (THRIVE) score [11]. These tools rely on readily available clinical features to estimate functional outcomes and are designed for use at patient admission. External validation of these scores has demonstrated moderate-to-high predictive accuracy, with AUC values ranging between 0.70 and 0.80 [12,13].

#### 4.2.1. Treatment-Specific Limitations of Existing Models

While these models are practical and convenient for predicting general functional outcomes, they have important limitations. Many do not account for specific treatment options or are tailored to a single intervention, such as TL (DRAGON score) or MT (THRIVE score), restricting their application across broader AIS patient populations, particularly when choosing between multiple treatment strategies.

In the internal validation, the ASTRAL, DRAGON, and THRIVE scores achieved AUCs of 0.85, 0.84, and 0.75, respectively. In comparison, the Stroke-SCORE not only demonstrated slightly better predictive accuracy, with an AUC of 0.86, but also offered a unique advantage: incorporating treatment-specific considerations. This makes the Stroke-SCORE a more comprehensive tool, capable of guiding individualized treatment decisions rather than merely predicting outcomes on the basis of baseline characteristics. However, while the Stroke-SCORE shows promise, external validation will be crucial to determine whether these advantages hold up across diverse populations and clinical settings.

#### 4.2.2. Population-Restricted Prediction Models

Many existing prediction models are limited in their applicability to specific patient subgroups. For example, the PREDICT score developed by Hoffmann et al. is designed for younger stroke patients (under 55 years) and uses features such as the ASPECTS (Alberta Stroke Program Early CT Score), plasma-glucose level, and large vessel occlusion type to predict 90-day outcomes. These specialized models lack the flexibility to be applied across a diverse range of AIS patients, making them less functional in general clinical practice [14].

Similarly, the MR PREDICTS tool represents a specialized model designed to identify patients who may benefit from intra-arterial treatment (IAT) for acute ischemic stroke. It incorporates nine predictors, including age, baseline NIHSS score, systolic blood pressure, history of ischemic stroke, diabetes mellitus, pre-stroke mRS score, ASPECTS, location of occlusion, and collateral score. MR PREDICTS demonstrated strong predictive performance with C-statistics of 0.79 (internal validation) and 0.73 (external validation), focusing on assessing good functional outcomes (mRS 0–2) at 90 days. However, its application is restricted to patients with CT-angiography (CTA)-diagnosed LVOs, making it particularly useful for specialists but less so for emergency physicians [15].

#### 4.2.3. Complex Models with Limited Real-Time Applicability

Other advanced models use complex metrics or specialized imaging biomarkers, making their incorporation into clinical workflows challenging, particularly in emergency settings where simplicity and speed are critical. For example, Forkert et al.’s model employed magnetic resonance fluid-attenuated inversion recovery (MR FLAIR) imaging to assess the final infarction volume and location to predict 30-day outcomes [16]. Despite its accuracy, this approach requires specialized imaging that may not always be feasible in real-time emergency scenarios. Similarly, the radiomics-based model developed by Yang and Guo relies on advanced imaging biomarkers that demand significant expertise and time-consuming preprocessing [17].

In contrast, the Stroke-SCORE offers a practical, treatment-specific prediction tool that relies on just three easily obtainable clinical characteristics: age, NIHSS score, and pre-mRS. This simplicity makes the Stroke-SCORE particularly suitable for real-time decision-making at the bedside, enabling clinicians to make quick yet informed treatment decisions based on individual risk profiles. However, while this simplicity is a key strength, the model provides only a static risk assessment at a single time point, failing to account for dynamic changes in patient condition over time. This limitation may reduce its applicability in scenarios where patient status evolves significantly during hospitalization.

### 4.3. Limitations and Future Directions

While internal validation has shown promising results, this study’s retrospective design and single-center dataset limit its generalizability; therefore, our results should be interpreted as hypothesis-generating rather than definite evidence. Although the simplicity of the model is a significant strength, its reliance on NIHSS and pre-mRS scores, which require evaluation by trained personnel, presents challenges in contexts where expertise is unavailable or inconsistent. Variability in these assessments, due to subjective interpretation, differences in training, or inconsistent application across evaluators, may introduce potential errors and impact the model’s predictive accuracy.

An additional limitation is the exclusion of 121 cases due to incomplete records, which could potentially introduce selection bias if the excluded patients had characteristics or outcomes differing from the analyzed cohort. Future efforts should aim to minimize data gaps to ensure a more representative analysis.

To address these limitations, we will first validate the Stroke-SCORE prospectively within our patient cohort to assess its performance in real-world scenarios. Future research should prioritize external, multicenter prospective validation to establish the Stroke-SCORE’s broader applicability, including its use in out-of-hospital settings such as pre-hospital emergency care. In these contexts, paramedics or first responders could benefit from tools like mobile applications or telemedicine platforms to perform rapid, standardized assessments of NIHSS and pre-mRS scores, enabling earlier risk stratification.

Additionally, integrating the Stroke-SCORE into electronic health record systems could streamline its application in clinical workflows, making it easier for clinicians to access and utilize. Within such systems, there is also potential to incorporate routinely measured bedside parameters, such as blood pressure or plasma-glucose, as additional predictors. These parameters are simple to obtain in acute settings and could enhance the model’s predictive accuracy while preserving its practical and efficient design.

With broader validation, the Stroke-SCORE has the potential to become a versatile and widely applicable tool, suitable for diverse clinical settings while maintaining its simplicity and ease of use.

## 5. Conclusions

The Stroke-SCORE is a practical, data-driven tool for personalizing AIS treatment. By utilizing three simple predictors—age, the NIHSS score, and the pre-mRS score—it balances predictive accuracy with clinical simplicity, making it suitable for real-world application. While promising for real-world application, the absence of external validation necessitates caution in interpreting these results as definitive.

By complementing existing treatment guidelines, the Stroke-SCORE has the potential to support informed, patient-centered decisions, improving outcomes and reducing unnecessary interventions. With further validation, it could become an essential tool in the management of acute ischemic stroke.

## Figures and Tables

**Figure 1 jpm-15-00018-f001:**
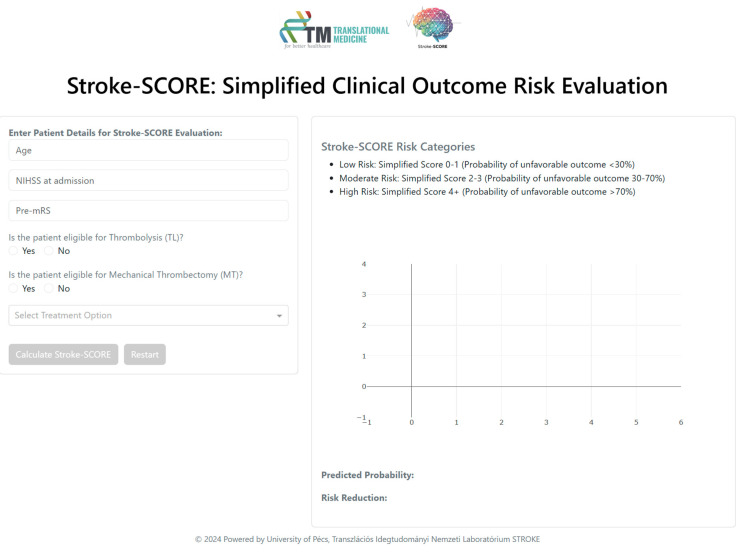
Stroke-SCORE interactive decision-support tool. Abbreviations: SCORE = Simplified Clinical Outcome Risk Evaluation; NIHSS = National Institutes of Health Stroke Scale; pre-mRS = pre-morbid modified Rankin Scale; TL = thrombolysis; MT = mechanical thrombectomy.

**Figure 2 jpm-15-00018-f002:**
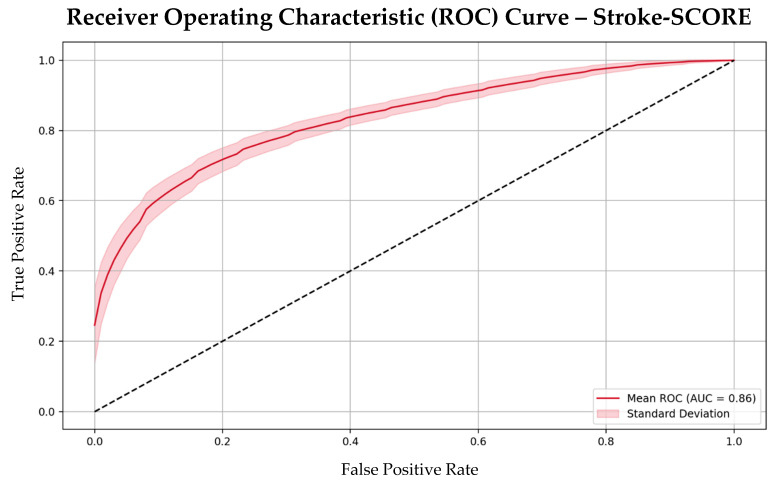
Receiver operating characteristic (ROC) curve with bootstrapping of the Stroke-SCORE. Abbreviations: SCORE = Simplified Clinical Outcome Risk Evaluation; AUC = area under the curve.

**Figure 3 jpm-15-00018-f003:**
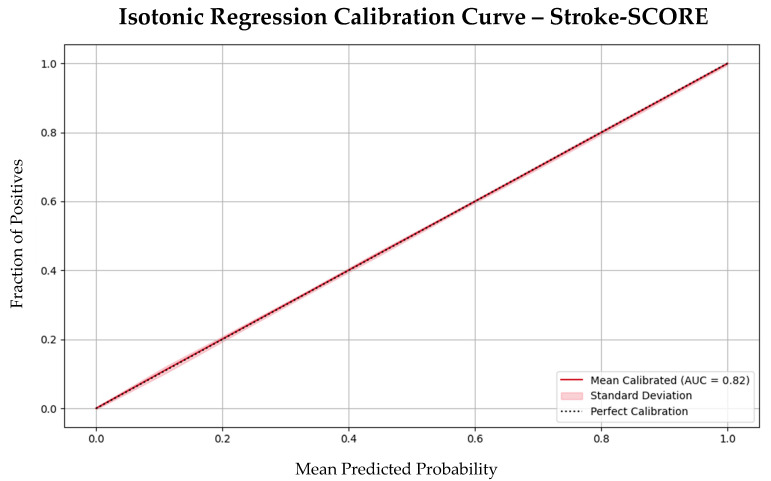
Isotonic regression calibration curve with bootstrapping of the Stroke-SCORE. Abbreviations: SCORE = Simplified Clinical Outcome Risk Evaluation; AUC = area under the curve.

**Figure 4 jpm-15-00018-f004:**
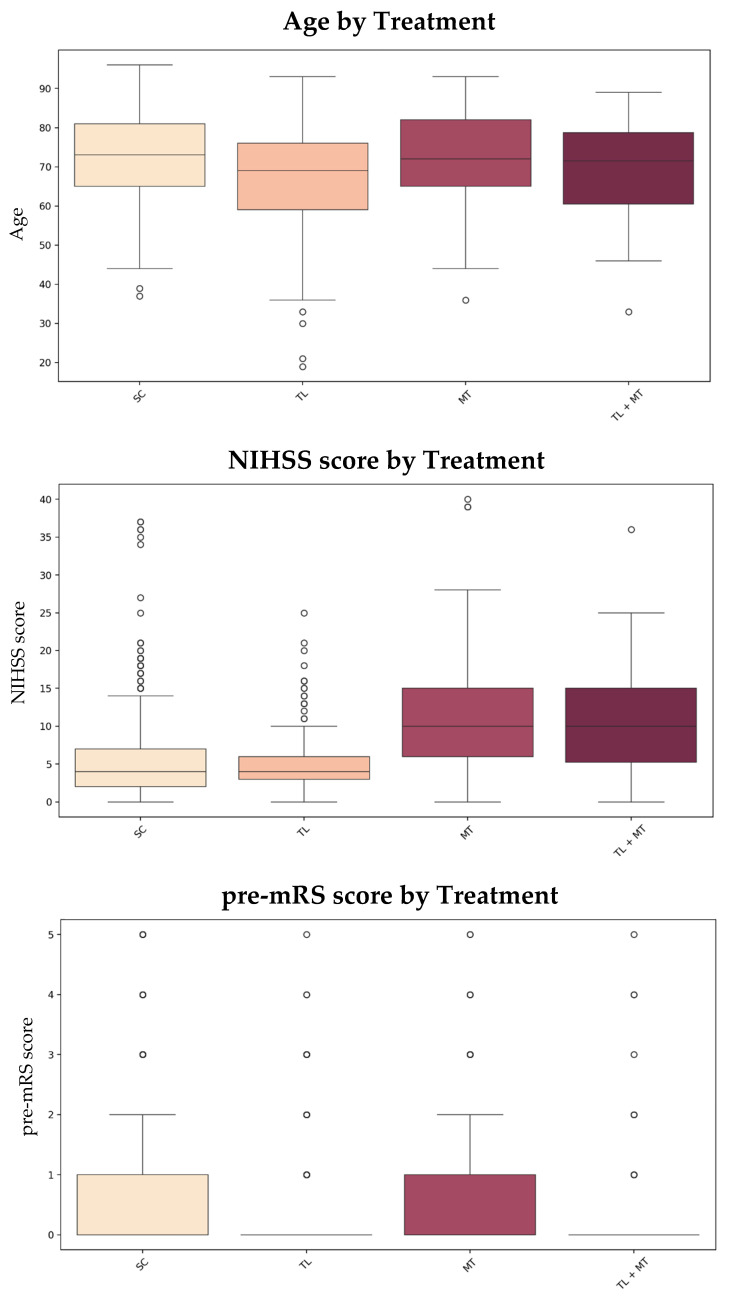
Box plots of age, NIHSS score, and pre-mRS score by treatment. Circles indicate outliers that fall outside the typical range for each treatment group. Abbreviations: SC = standard care; TL = thrombolysis; MT = mechanical thrombectomy; NIHSS = National Institutes of Health Stroke Scale; pre-mRS = pre-morbid modified Rankin Scale.

**Figure 5 jpm-15-00018-f005:**
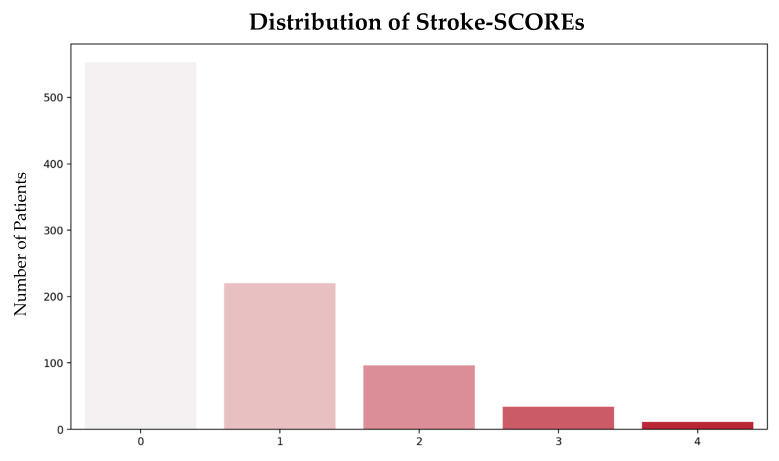
Distribution of Stroke-SCOREs. Abbreviations: SCORE = Simplified Clinical Outcome Risk Evaluation.

**Figure 6 jpm-15-00018-f006:**
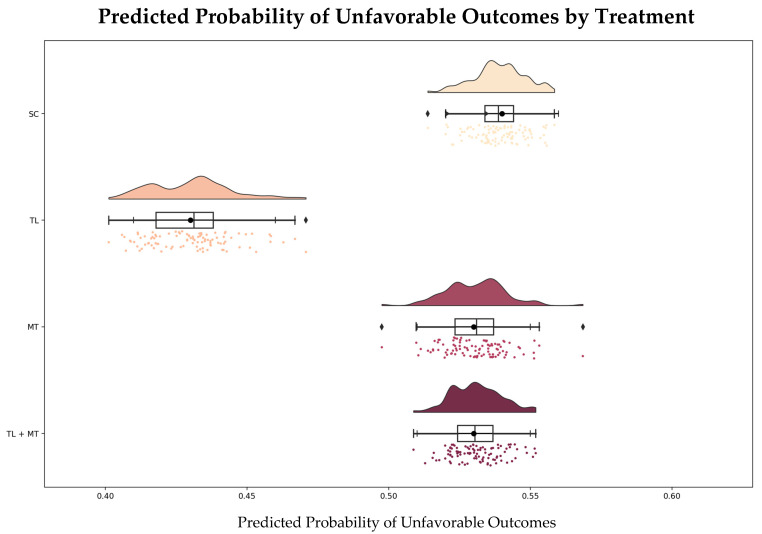
Predicted probability of unfavorable outcomes by treatment. Dots: Individual predicted probabilities. Boxplots: Show the interquartile range (IQR) with the median (black line inside the box) and whiskers extending to 1.5 times the IQR. Black dots: Mean predicted probability for each treatment. Horizontal Error Bars: Indicate 95% confidence intervals around the mean. Diamonds (if present): Represent outliers that fall outside the whiskers. Abbreviations: SC = standard care; TL = thrombolysis; MT = mechanical thrombectomy.

**Table 1 jpm-15-00018-t001:** Predictors of unfavorable outcomes at 90 days.

Feature	Logistic Regression Coefficient	OR [95% CI] and *p*-Value	Mean SHAP Value	95% CI for Mean SHAP Value	Percentage Importance
Age	0.0507	OR = 1.05 [0.34–0.67], *p* < 0.001	0.5114	0.0670, 0.2136	17.39%
NIHSS score	0.1710	OR = 1.19 [0.13–0.21], *p* < 0.001	0.7039	−0.0283, 0.1770	23.94%
Pre-mRS score	0.9666	OR = 2.63 [0.73–1.21] *p* < 0.001	0.8887	−0.1258, 0.1480	30.22%
Plasma-glucose	0.0672	OR = 1.07 [0.02–0.13], *p* = 0.044	0.2478	−0.0456, 0.0385	8.42%
INR	0.1591	OR = 1.17 [−0.30–0.62], *p* = 0.498	0.1625	−0.0276, 0.0313	5.53%
Onset–door time	∞	OR = 1.00, *p* = 0.753	0.3080	−0.0146, 0.0964	10.47%
HT	0.1319	OR = 1.14 [−0.36–0.63], *p* = 0.600	0.0325	−0.0063, 0.0052	1.11%
DM	0.3951	OR = 1.48 [−0.02–0.81], *p* = 0.060	0.0857	−0.0113, 0.0145	2.92%

Abbreviations: OR = odds ratio; CI = confidence interval; SHAP = SHapley Additive exPlanations; NIHSS = National Institutes of Health Stroke Scale; pre-mRS = pre-morbid modified Rankin Scale; INR = international normalized ratio; HT = hypertension; DM = diabetes mellitus.

**Table 2 jpm-15-00018-t002:** Average 90-day mRS by treatment.

Stroke-SCORE	SC	95% CI	TL	95% CI	MT	95% CI	TL + MT	95% CI
0	1.71	1.46–1.96	1.10	0.84–1.36	2.68	2.13–3.22	1.76	1.05–2.48
1	3.38	3.00–3.76	2.74	2.00–3.47	3.82	3.19–4.46	3.31	2.17–4.45
2	4.91	4.50–5.33	3.29	1.04–5.53	3.90	3.24–4.56	3.40	1.81–4.99
3	5.54	5.01–6.07	4.50	0.00–6.00	5.30	4.23–6.00	5.83	5.40–6.00
4	6.00	6.00–6.00	N/A	-	5.67	4.23–6.00	N/A	-

Abbreviations: mRS = modified Rankin Scale; SCORE = Simplified Clinical Outcome Risk Evaluation; SC = standard care; CI = confidence interval; TL = thrombolysis; MT = mechanical thrombectomy; N/A = not applicable.

## Data Availability

The original contributions presented in the study are included in the article and further inquiries can be directed to the corresponding author.

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
