# Peer review of "Stroke-SCORE: Personalizing Acute Ischemic Stroke Treatment to Improve Patient Outcomes"

_jpm, 2025, doi:10.3390/jpm15010018_

Round 1
Reviewer 1 Report
Comments and Suggestions for Authors
Thank you for the opportunity to review the manuscript titled "Stroke-SCORE: Personalizing Acute Ischemic Stroke Treatment to Improve Patients Outcomes" I found many strengths in the study, but there are some issues that must be addressed before the manuscript can be accepted for publication:
Introduction:
· The authors should further justify the clinical relevance of the problem using epidemiological data.
· The authors should better explain the specific deficiencies of the current protocols, which more strongly justify the development of Stroke-Score.
· The introduction is too short, giving a very limited view of the problem situation.
Material and methods:
· The authors adequately explain the process of selecting patient records. There are 13% of cases that the authors could not take into account because they were not complete, which may induce unanalysed bias.
· The model does not include variables that could have improved the predictive accuracy of the scale, such as the patient's comorbidities or the situation in which the onset of symptoms occurred. The authors should explain the reason for not having taken them into account.
· The lack of external validation of the model significantly limits the results. It is recommended, to the extent possible, to carry out it.
· The authors do not specify how the quality of the data collected in the registry was controlled, an essential aspect in a retrospective study.
Results:
· The authors should present the specific results by risk groups in more detail.
· The subgroup analyses and simulations of results are very interesting, although they are not supported by additional statistical analyses.
· In the presentation of the results, the confidence intervals for the predictions and the results of the subgroups are missing.
· The authors do not analyse how the characteristics of the patients may have influenced the results of the model.
Discussion:
· The authors should begin this section by indicating the main results obtained in the study.
· The comparison with other existing models is very superficial, lacking a critical analysis of the disadvantages of the Stroke-Score model compared to these tools.
· The practical implications of implementing this model in different clinical contexts, such as the out-of-hospital setting or those with limited resources, are not sufficiently explored.
· Although the authors propose future validations, specific adjustments that could improve model performance are not discussed.
· The authors do not sufficiently discuss the possible sources of error in the measurements made (subjectivity in the evaluation of the NIHSS or pre-mRS
· The authors have not taken into account the impact of patients excluded due to missing data on the final results
Conclusion:
· The conclusions are overly optimistic, as the study does not include any external validation or thorough analysis of practical limitations.
· The authors should further discuss the next steps needed to make this tool clinically viable.
Reviewer 2 Report
Comments and Suggestions for Authors
Seetge et al. present an interesting manuscript on the utility of Stroke-Score (Simplified Clinical Outcome Risk Evaluation) as a tool to predict AIS outcomes. They based their analysis on a retrospective cohort of 793 patients with AIS. The main conclusion of this paper is that Stroke-Score could serve as a practical tool to predict AIS outcome and assist clinicians to make more personalized decision in AIS care.
Here are my comments on this paper:
In the methodology, the definition of AIS should be provided and properly cited. Group selection should be explained in more detail (e.g. exclusion due to incomplete data, were only patients with complete data included?) Metrics measured in the study should also be listed and scales analyzed should be cited. Telephone interview form should be translated into English and included in the Supplementary Appendix. Description of Stroke SCORE should be expanded, it is the most important part of this paper, what factors it analyzes, how it works, etc.
In addition, it is controversial to define unrecoverable outcome based only on mRS score, patients with preexisting disability will always be discriminated in such model. Furthermore, the strongest predictors of unfavorable outcome were more severe stroke, age and disability. Nevertheless, it is known that these patients despite higher risk and lower perspective for improvement can benefit from treatment (what is reflected in stroke guidelines). I wonder if analysis of more uniform cohorts (e.g. only treated with thrombolysis or only without preexisting disability) would be better to validate this tool. This should at least be discussed in the limitations.
Minor comments:
Line 52 – comma is missing
Line 54 – patiens medical data were analysed
All abbreviation in figures need to be describe in figures description
Reviewer 3 Report
Comments and Suggestions for Authors
Thank you for the opportunity to reiew your interesting and well written study.
Just one revision suggested to consider adding to the discussion https://www.mrclean-trial.org/mr-predicts.html and put your study in the context of that calculator.
Round 2
Reviewer 1 Report
Comments and Suggestions for Authors
Dear Editor and authors,
Although most of the comments and suggestions have been addressed, those concerning the methodology and results should still be taken into account before considering the manuscript for publication.
Best regards
Reviewer 2 Report
Comments and Suggestions for Authors
The manuscript was modified according to my sugestions.
